# TRIM34 restricts HIV-1 and SIV capsids in a TRIM5α-dependent manner

**Molly Ohainle**[1]*, **Kyusik Kim**[2◉], **Sevnur Komurlu Keceli**[3◉], **Abby Felton**[1],
**Ed Campbell**[3], **Jeremy Luban**[2], **Michael Emerman**[1]*

**1** Divisions of Human Biology and Basic Sciences, Fred Hutch, Seattle, Washington, United States of America, **2** Program in Molecular Medicine, University of Massachusetts Medical School, Worcester, Massachusetts, United States of America, **3** Department of Microbiology and Immunology, Stritch School of Medicine, Loyola University, Chicago, Maywood, Illinois, United States of America

◉ These authors contributed equally to this work.
* mohainle@fredhutch.org (MO); memerman@fredhutch.org (ME)

**Data Availability Statement:** All relevant data are within the manuscript, its Supporting Information files or available through the Gene Expression Omnibus (GEO - GSE140467).

## Abstract

The HIV-1 capsid protein makes up the core of the virion and plays a critical role in early steps of HIV replication. Due to its exposure in the cytoplasm after entry, HIV capsid is a target for host cell factors that act directly to block infection such as TRIM5α and MxB. Several host proteins also play a role in facilitating infection, including in the protection of HIV-1 capsid from recognition by host cell restriction factors. Through an unbiased screening approach, called HIV-CRISPR, we show that the CPSF6-binding deficient, N74D HIV-1 capsid mutant is sensitive to restriction mediated by human TRIM34, a close paralog of the well-characterized HIV restriction factor TRIM5α. This restriction occurs at the step of reverse transcription, is independent of interferon stimulation, and limits HIV-1 infection in key target cells of HIV infection including CD4+ T cells and monocyte-derived dendritic cells. TRIM34 can also restrict some SIV capsids. TRIM34 restriction requires TRIM5α as knockout or knockdown of TRIM5α results in a loss of antiviral activity. Through immunofluorescence studies, we show that TRIM34 and TRIM5α colocalize to cytoplasmic bodies and are more frequently observed to be associated with infecting N74D capsids than with WT HIV-1 capsids. Our results identify TRIM34 as an HIV-1 CA-targeting restriction factor and highlight the potential role for heteromultimeric TRIM interactions in contributing to restriction of HIV-1 infection in human cells.

## Author summary

HIV-1 infection in humans is the result of cross-species transmission of a related primate lentivirus from chimpanzees. In order to adapt to replicate in human cells, HIV-1 adapted to host proteins that restrict infection by viruses. The HIV-1 capsid protein encapsidates the HIV-1 genome, allowing it to be delivered to the host cell for integration into the host chromosome. However, on delivery to the cytoplasm the HIV-1 capsid protein is exposed to an array of capsid-targeting restriction factors that act to limit infection. Viral mutants can be used to reveal some of these restriction factors. Our results show that TRIM34 is an

**Funding:** This work was supported by grants from National Institutes of Health (NIH): R01 AI147877 (M.O.), CFAR AI027757 (M.O.), P50 AI150464-13 (M.O.), R01 AI30927 (M.E), R37 AI147868 (J.L.) and R01 AI093258 (E.C.). The funders had no role in study design, data collection and analysis, decision to publish, or preparation of the manuscript.

**Competing interests:** The authors have declared that no competing interests exist.

antiviral effector that blocks infection of an HIV with a particular mutation in capsid. TRIM34 also inhibits some lentiviruses that infect monkeys. This block requires the closely-related paralog TRIM5α, a capsid-targeting restriction factor with well-characterized antiviral activity in primates. These results highlight the complex interaction between HIV-1 with human antiviral proteins.

## Introduction

The HIV-1 capsid protein (CA or p24$^{gag}$) forms the core of the virion and is key to effective delivery of the HIV-1 genome inside a host cell and into the nucleus where integration into the host chromosome occurs [1, 2]. HIV-1 CA is involved in the early steps of HIV-1 replication including uncoating, nuclear entry, and integration site selection [1–4]. HIV-1 CA is also an important target for host restriction factors such as rhesus TRIM5α (the alpha isoform of the *TRIM5* gene) and MXB (reviewed in [1, 5]).

Many restriction factors are induced by type I Interferon (IFN). Although Human TRIM5α, previously thought to lack significant activity against primate lentiviruses, has recently been shown to restrict wild type (WT) HIV-1 capsids in IFN-treated cells [6, 7]. TRIM5α restriction of retroviral capsids is driven by interactions between the C-terminal SPRY domain of TRIM5α and determinants present in assembled CA structures [8]. While the affinity of the SPRY domain for CA is low, this low affinity is overcome by TRIM5α dimerization and its ability to form higher-order assemblies around the viral core, enhancing avidity of the TRIM5α-CA interaction [8]. TRIM5α is also able to oligomerize with other TRIM-family members [9, 10]. One key aspect of TRIM biology that remains relatively unexplored includes the potential for hetero-oligomerization of TRIM proteins that could have important functional consequences.

Through the study of HIV-1 CA mutants that lack binding to host cell factors or possess other key phenotypes, such as altered stability, much has been revealed about how CA determines the fate of HIV-1 cores inside cells. For example, the host proteins CPSF6 and Cyclophilin A (CypA) have a complex but important role in HIV-1 CA interactions and infection [1]. HIV-1 CA binds CypA which provides protection against the action of TRIM5α [11, 12]. CPSF6 interacts with HIV-1 capsid on entry into target cells [13, 14] and facilitates interaction with nuclear import pathways that enhances targeting of HIV-1 integration into gene-rich regions [15, 16]. Single amino acid mutations in the HIV-1 capsid protein, for example N74D for CPSF6 and P90A for CypA, abrogate binding to these host factors [13, 17]. Both capsid mutants have been demonstrated to infect cells less efficiently than wild type (WT) in some cell types, including primary cell such as CD4+ T cells and monocyte-derived macrophages (MDMs) [12, 17–19]. Further, both the HIV-1 P90A capsid mutant and the HIV-1 N74D capsid mutant, referred to hereafter as P90A and N74D respectively, have been shown to be hypersensitive to the effects of IFN [19], suggesting that one or more IFN-induced restriction factors block infection of these capsid mutant viruses. Restriction of these mutants has been shown to be independent of the IFN-induced capsid-targeting restriction factor MxB [19] but identification of other capsid-targeting restriction factors underlying the increased IFN sensitivity of these CA mutants has been elusive.

Previously, we demonstrated that human genes that mediate the antiviral effects of IFN can be identified through an unbiased CRISPR screening approach called HIV-CRISPR [6]. Here we use this approach to identify capsid-targeting restrictions that target the P90A and N74D HIV-1 capsid mutants. While the CypA-binding deficient P90A mutant becomes more

sensitive to TRIM5α restriction, the CPSF6-binding deficient N74D mutant becomes sensitive to a novel restriction by the TRIM5α paralog, TRIM34. This restriction is independent of IFN induction as well as CPSF6 binding and results in a block during HIV reverse transcription. TRIM34 restriction occurs in primary cells in addition to the THP-1 monocytic cell line used in our screens. Further, we find that TRIM34 requires TRIM5α to inhibit N74D while inhibition of P90A occurs independent of TRIM34. Thus, we find that TRIM34 is a novel inhibitor of HIV-1 and SIV capsids that acts in conjunction with TRIM5α to limit infection of primary T cells.

## Results

### HIV-CRISPR screening identifies TRIM34 as an inhibitor of the HIV-1 N74D capsid mutant

P90A and N74D have been shown to be impaired in replication both in IFN-treated and untreated cells [17–20]. Therefore, we hypothesized that the P90A (CypA-deficient) and N74D (CPSF6-deficient) capsid mutants may be more sensitive to inhibition by capsid-targeting restriction factors in human cells. Two possible outcomes are that these mutants are either more sensitive to the same restrictions that target wild type capsids or that they are sensitive to novel capsid-targeting restriction factor(s).

To identify the host cell restrictions targeting these capsid mutant viruses, we used our unbiased screening approach, HIV-CRISPR screening, to ask what genes in our library of ~2000 genes enriched in Interferon-Stimulated Genes (ISGs) [6] are responsible for inhibiting both mutants in THP-1 cells. HIV-CRISPR screening is a virus-packageable CRISPR screening approach in which infecting HIV virions package the HIV-CRISPR modified lentiviral vector *in trans* upon budding from the infected cell [6]. As the level of virus replication is dependent on the phenotype of gene knockout introduced by Cas9 endonuclease and sgRNA encoded in the HIV-CRISPR vector, the virus itself serves to readout the barcodes of gene knockouts with effects on virus replication (Fig 1A). Quantification of individual 20bp sgRNA sequences enriched in the virions relative to the representation of sgRNA sequences in the cell populations allows for the identification of antiviral genes as gene knockouts that allow for more robust replication of each virus. In the HIV-CRISPR screens described here, a library of cells transduced with the HIV-CRISPR vector targeting genes enriched in ISGs, the PIKA$_{HIV}$ library [6], was infected with WT, N74D or P90A viruses after overnight treatment with IFN alpha.

Cell pellets and viral supernatants were collected and both genomic DNA from cells and viral RNA from virions were isolated and amplified for deep sequencing and MAGeCK analysis [21] to identify sgRNA sequences significantly enriched in the viral supernatant of each virus (Fig 1B: WT vs P90A; Fig 1C: WT vs N74D; see S1 Table for full screen results). The four genes in the library that are essential for IFN signaling, *STAT1*, *STAT2*, *IFNAR1* and *IRF9*, were the highest-scoring hits in both screens as knockout of any gene in this pathway results in rescue of IFN inhibition of viral replication (Fig 1B and 1C: magenta). Next, a core set of ISGs, including *MXB (MX2)*, *IFITM1*, *TETHERIN (BST2)* and *TRIM5*, were some of the highest-scoring gene hits for all three viruses screened (WT, N74D and P90A viruses). Therefore, these restriction factors target the capsid mutants similar to what we previously found for wild type HIV-1 [6] and as we observed again here (Fig 1B and 1C: cyan, yellow and dark blue, respectively). In both capsid mutant screens we measured a lower rank for *MXB* than for WT (Fig 1B and 1C–compare rank in WT screen as compared to either capsid mutant), consistent with the previously-reported relative resistance of both P90A and N74D to restriction by MXB [19, 22–27]. In contrast, *TRIM5* is the highest-scoring gene hit for P90A (Fig 1B: yellow). This is

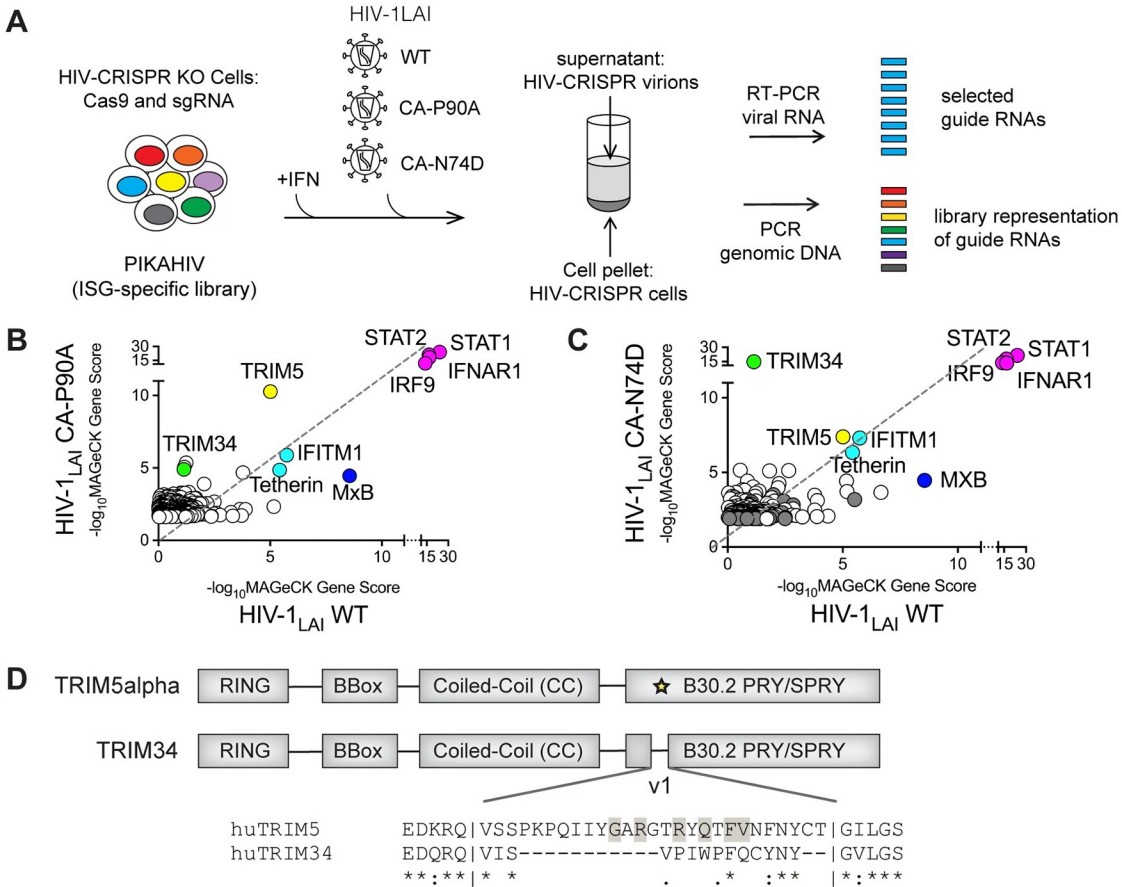

**Fig 1. HIV-CRISPR screening identifies TRIM5α and TRIM34 as restriction factors for HIV-1 capsid mutant viruses, P90A and N74D.** A: HIV-CRISPR screens were performed after overnight IFN induction in ZAP-KO THP-1 cells transduced with the PIKA$_{HIV}$ (ISG-specific) library. HIV-CRISPR cells were infected with either WT HIV-1$_{LAI}$, the HIV-1 N74D capsid mutant (HIV-1$_{LAI}$ CA-N74D) or the HIV-1 P90A capsid mutant (HIV-1$_{LAI}$ CA-P90A). All of these viruses are variants of an HIV-1$_{LAI}$ provirus with GFP cloned in place of Nef (pBru3GFP3). B and C: Two replicates of the WT and N74D screens (C) were performed while the P90A screen (B) includes only a single replicate. At 3 days post-infection cells and viral supernatants were collected, genomic DNA and viral RNA was extracted and 20bp sgRNA cassettes amplified by PCR or RT-PCR, respectively. MAGeCK Analysis of the enrichment of 20bp sgRNA sequences in viral RNA as compared to the genomic DNA was performed to calculate a MAGeCK Gene Score. Magenta: IFN pathway genes. Cyan: Gene hits shared across screens. Green: TRIM34. Yellow: TRIM5α. Dark Blue: MxB. Gray: Non-Targeting Controls (NTCs). X-Axis: inverse log MAGeCK Gene Score (-log$_{10}$MAGeCK Gene Score) for the WT HIV-1 screen. Y-Axis: inverse log MAGeCK Gene Score (-log$_{10}$MAGeCK Gene Score) for the HIV-1 P90A CA mutant screen (B) or the HIV-1 N74D CA mutant screen (C). The Top 200 Gene hits for each capsid mutant virus (CA-P90A or CA-N74D) along with corresponding scores for the WT virus are shown (the same data is used for WT in both B and C). D: A schematic of the human TRIM5α and human TRIM34 domain structures, including the RING, B-Box, Coiled-Coil and B30.2 PRY/SPRY domains. An alignment of the v1 region (yellow star) of the SPRY domains of human TRIM5α and human TRIM34 is shown below the protein schematic. Identical residues are shown with an asterisk, similarity is indicated with a colon. Gray shading indicates residues in the v1 region identified to be evolving under positive selection in primate *TRIM5* by Sawyer et al [31].

consistent with recent results showing that loss of CypA binding by HIV-1 capsids results in sensitivity to TRIM5α restriction [12].

For the N74D screen, we find a novel HIV-1 restriction factor gene, *TRIM34*, as the highest-scoring hit that is not found in the screen with WT HIV-1 (Fig 1C: Green). *TRIM34* scores as highly as the IFN pathway genes (Fig 1C: Magenta), highlighting the key role TRIM34 potentially plays in blocking replication of the N74D capsid mutant virus. We also find *TRIM34* as a less significant hit in the P90A screen as compared to the N74D screen,

suggesting it may have a minor role in restriction of the P90A virus at least in some cell types (Fig 1B). TRIM34 was first described to be an Interferon-Stimulated Gene in HeLa cells [28] and is in a cluster of paralogous human *TRIM* genes that includes *TRIM5*, *TRIM22* and *TRIM6* on human chromosome 11 [29]. Like all members of the *TRIM* gene family [8], *TRIM34* encodes an N-terminal RBCC or tripartite motif, including RING, B-Box and coiled-coil domains (Fig 1D) [28]. TRIM34 shares 56% amino acid identity and overall domain structure with the closely-related, capsid-targeting HIV-1 restriction factor TRIM5α (Fig 1D). Therefore, TRIM34 could share some functional features of TRIM5α biology. However, unlike TRIM5α, TRIM34 has some surprising features. First, *TRIM34* is not a significantly rapidly-evolving gene in primates [29, 30]. Positive selection is frequently a feature of host antiviral genes in longstanding conflict with pathogens; therefore, the lack of positive selection in *TRIM34* is unexpected. Further, human *TRIM34* has a deletion in the v1 region of the B30.2 PRY/SPRY domain (Fig 1D). The v1 loop of the TRIM5α SPRY domain contains multiple sites of positive selection in primate *TRIM5* and is known to be important for determining the specificity of capsid recognition by TRIM5α [31]. These findings suggest that while TRIM34 may have some structural and functional homology with TRIM5α, TRIM34 restriction of this HIV mutant is also likely to differ in some ways from TRIM5α.

## TRIM34 inhibits N74D at a step before completion of reverse transcription

To validate the screen results (Fig 1) which suggest that TRIM34 restricts N74D but not WT HIV-1, we knocked out TRIM34 in THP-1 cells by transduction with a lentiviral vector encoding Cas9 and two different *TRIM34*-specific sgRNAs together with Non-Targeting Control (NTC) sgRNAs (Fig 2A and 2B). Efficient *TRIM34* gene knockout was confirmed by sequencing analysis of the *TRIM34* locus in both populations of cells (Fig 2A and 2B–63% edited alleles for TRIM34KO_1 and 83% edited alleles for TRIM34KO_2). Control and TRIM34-KO cell populations were then infected with both WT and N74D viruses after overnight IFN treatment and the % infected cells was assayed by flow cytometry. We observed no significant effect of TRIM34 knockout on the replication of the WT virus (Fig 2A; *P*>0.05). In contrast, we observe 2.2-fold rescue of N74D replication across both *TRIM34* KO pools (Fig 2A; *P<0.001*), consistent with our HIV-CRISPR screen results that found *TRIM34* as a block to N74D infection in THP-1 cells.

To ask if TRIM34 restriction occurs only in IFN-treated cells as has recently been shown for TRIM5α [7], we also infected control and TRIM34-KO cells with WT and N74D virus without any IFN treatment and assayed the % infected cells by flow cytometry (Fig 2B). Unlike TRIM5α inhibition of wild type HIV-1 capsids [7], TRIM34 is a constitutive inhibitor of the N74D CA mutant as KO rescues infection of the N74D virus even in the absence of IFN treatment 2.0-Fold (Fig 2B; *P<0.001*). Further confirming these findings, we repeated the HIV-CRISPR screen in THP-1 cells without any IFN treatment and still find *TRIM34* as a hit that is specific to the HIV-1 N74D virus as *TRIM34* is the 2[nd] highest ranked gene in this screen without IFN (Fig 2C–black bar). Importantly, while we initially identified TRIM34 in a screen for IFN-induced genes that block the N74D virus, these data suggest that TRIM34 is actually a constitutive block to infection whose activity is IFN-independent. Therefore, subsequent experiments are performed in the absence of IFN.

We also determined whether or not TRIM34 plays a role in restriction of HIV-1 capsids in primary activated, CD4+ T cells. *TRIM34* KO CD4+ T cell pools were generated through electroporation of CD3/CD28-activated, CD4+ T cells with CRISPR/Cas9 complexes targeting *TRIM34* or Non-Targeting Control complexes. 54% of *TRIM34* alleles were found to be edited in this experiment (ICE KO-Score). We infected the control and TRIM34-KO pools with both

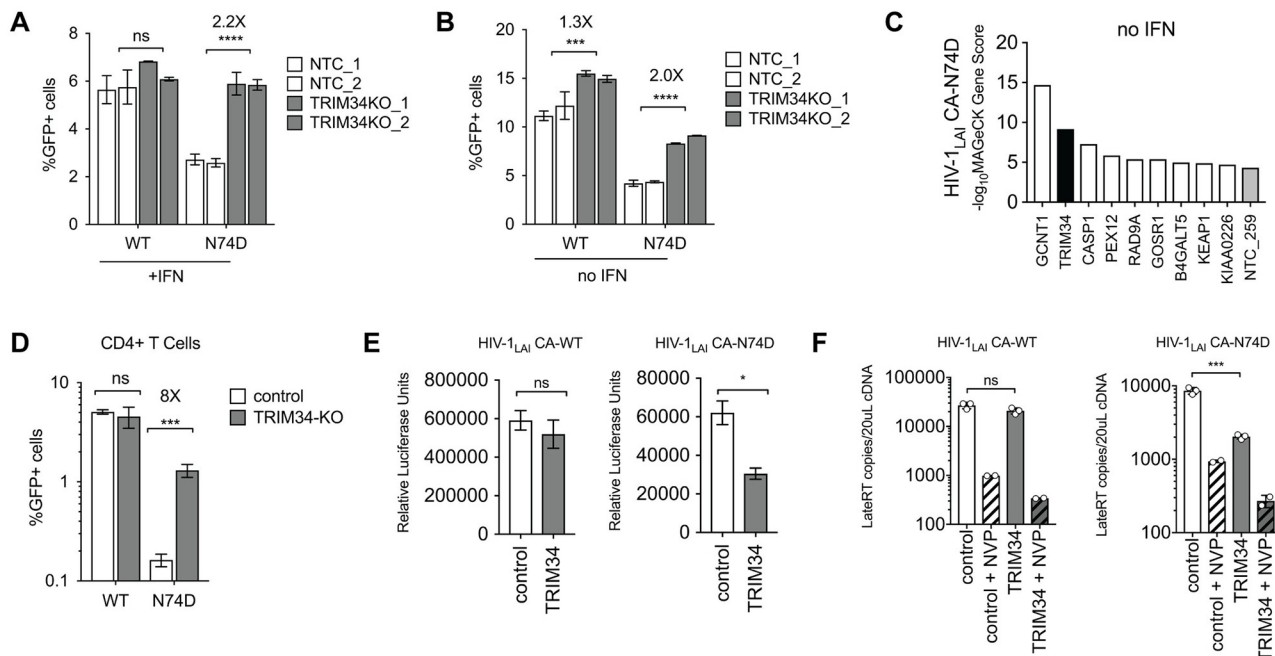

**Fig 2. TRIM34 blocks HIV replication at the reverse transcription step in THP-1 and primary CD4+ T cells independent of IFN treatment.** A and B: THP-1 cells were transduced with a lentiviral vector encoding Cas9 and an sgRNA targeting either TRIM34 (Gray Bars: $n$ = 2 pools, each an independent sgRNA) or a Non-Targeting Control (NTC) sgRNA (White Bars: $n$ = 2 pools, each an independent sgRNA). TRIM34 KO in each cell pool was determined by ICE Analysis (TRIM34-KO_1 = 63% ICE KO-Score, TRIM34_2 = 83% ICE KO-Score). Edited cells were infected with either WT or N74D HIV and the amount of HIV replication assayed by analysis of GFP+ cells 2 days post-infection after overnight IFN treatment (A) or in untreated cells (no IFN) (B). $P$ values were calculated by combining all replicates from triplicate infections in both edited populations for either the NTC control or TRIM34 KO. The fold increase in infection in the TRIM34-KO cells as compared to the NTC cells is shown for each virus in each condition. Data are represented as the mean +/- s.d. C: An HIV-CRISPR screen (with the ISG-enriched PIKA$_{HIV}$ library) was performed with a single replicate in THP-1 cells without any IFN treatment for the N74D capsid mutant virus. MAGeCK Gene Scores (-log$_{10}$MAGeCK Gene Scores) for the Top 10 gene hits are shown. Black bar: TRIM34. Gray bar: Non-Targeting Control (NTC). White bars: other gene hits. D: Primary, activated CD4+ T cells were electroporated with Cas9-RNP complexes targeting TRIM34 (gray bars) or control complexes (white bars: control). TRIM34 KO in this pool was determined to be edited at 54% (ICE KO-Score). 2 days post-electroporation edited CD4+ T cell pools were infected with GFP reporter HIV-1 viruses (WT or N74D) and infection levels assayed 2 days later by flow cytometry. Data are represented as the mean +/- s.d. from triplicate infections. E: THP-1 cells were transduced with a lentiviral vector (pHIV-dTomato) encoding TRIM34 (gray bars) or empty vector (white bars: control). Cell populations were sorted for dTomato expression and, following recovery, were infected with VSV-G pseudotyped WT or N74D Luciferase reporter viruses (HIV-1$_{LAI}$ CA-WT LUC or HIV-1$_{LAI}$ CA-N74D LUC). Two days later levels of infection were assayed through a Luciferase Assay. Data are represented as the mean +/- s.d. from duplicate infections. F: TRIM34-overexpressing cells (gray bars) or control cells (white bars) were infected with VSV/G-pseudotyped HIV-1$_{LAI}$ CA-WT LUC or HIV-1$_{LAI}$ CA-N74D LUC viruses with or without Nevirapine (NVP) to inhibit HIV reverse transcription. 16 hours later viral cDNA was collected and levels of HIV reverse transcription products were assayed by qPCR. Data are represented as the mean +/- s.d. from triplicate (no NVP) or duplicate (+NVP) infections. $P$ values were determined using two-sided unpaired $t$-tests (ns = not significant, $^{*}P<0.05$, $^{**}P<0.01$, $^{***}P<0.001$, $^{****}P<0.0001$).

WT and N74D viruses and measured infection levels after 2 days by flow cytometry. We do not observe any rescue of WT infection on knockout of TRIM34 (Fig 2D; ns). However, knockout of TRIM34 in primary CD4+ T cells has an 8-fold effect which is an even stronger effect on N74D than what we observed in THP-1 cells (Fig 2D; $P<0.01$, compare with Fig 2A). Therefore, endogenously-expressed TRIM34 is a constitutive HIV-1 restriction factor in these key HIV-1 target cells.

TRIM5α binds to and blocks HIV-1 capsids in the cytoplasm, resulting in significantly decreased reverse transcription products during infection [8]. Moreover, there is a block prior to reverse transcription of the N74D CPSF6-binding capsid mutant in macrophages [18]. Thus, we hypothesized that TRIM34 would mediate a similar block before reverse transcription would occur. In order to test this hypothesis, we stably-overexpressed human TRIM34 in clonal TRIM34-KO THP-1 cells and assayed replication of both the wild type and the N74D

CA mutant viruses. TRIM34 overexpression does not result in any significant inhibition of WT HIV-1 (Fig 2E, left panel). In contrast, we observe significant inhibition of the N74D mutant by TRIM34 (Fig 2E, right panel). Therefore, as predicted for a restriction factor, over-expression of TRIM34 in THP-1 inhibits infection of cells by the HIV-1 N74D CA mutant virus. To ask if TRIM34 blocks infection before reverse transcription, similar to rhesus TRIM5α, we infected control and TRIM34-overexpressing THP-1 cells and assayed viral DNA accumulation through a qPCR assay that detects HIV-1 reverse transcription products [32]. Infections in the presence of Nevirapine (NVP) were performed in parallel to determine back-ground levels of DNA from viral stocks (Fig 2F). The inhibition of replication of the HIV-1 N74D CA virus (Fig 2E) in TRIM34-overexpressing cells is correlated with a similar decrease in the accumulation of HIV viral DNA (Fig 2F). Therefore, TRIM34, like its paralog TRIM5α, inhibits HIV-1 replication early in the viral life cycle before the completion of reverse transcription.

## TRIM34 restricts HIV-1 and SIV capsids independent of CPSF6 binding

The N74D CA mutant virus was first characterized due to its loss of CPSF6 binding [13]. Therefore, we reasoned that loss of CPSF6 binding could expose HIV-1 CA to restriction by TRIM34. To ask if the loss of CPSF6 binding is sufficient to sensitize HIV-1 capsids to TRIM34 restriction, we tested another CPSF6-binding capsid mutant HIV-1 A77V that, like HIV-1 N74D, also results in loss of binding to CPSF6 [33]. In contrast to infection with HIV-1 N74D CA, we find that infection of TRIM34-overexpressing cells is equivalent to WT cells for the HIV-1 A77V mutant (Fig 3A). We further tested the relative sensitivity of HIV-1 N74D and HIV-1 A77V in primary cells by knocking out *TRIM34* in primary CD4+ T cells (Fig 3B– TRIM34-KO 52% ICE KO-score). We measured infection of these *TRIM34*-KO cells by the HIV-1 N74D and HIV-1 A77V CA mutants as compared to a WT HIV-1 control (Fig 3B). As shown in Fig 2, infection with the N74D CA mutant can be rescued by TRIM34 knockout in CD4+ T cells (Fig 3B). Consistent with our overexpression assay in THP-1 cells we find that the HIV-1 A77V mutant is not rescued by TRIM34 knockout in CD4+ T cells (Fig 3B). There-fore, in both THP-1 cells and primary CD4+ T cells the HIV-1 A77V mutant that lacks binding to CPSF6 does not become sensitive to TRIM34 restriction. To further explore the CPSF6-in-dependence of TRIM34 restriction, we also assayed replication of another CPSF6-binding defi-cient mutant, N57A, in our ectopic overexpression system. In contrast to the restriction of the N74D capsid mutant virus measured in TRIM34-overexpressing cells, we do not observe any restriction of the N57A capsid mutant (Fig 3C). Therefore, restriction by TRIM34 is indepen-dent of CPSF6 binding status of the HIV-1 capsid. Instead restriction of the HIV-1 N74D CA by TRIM34 is determined by a feature of this capsid other than CPSF6 binding.

Restriction factors from one species often potently restrict primate lentiviruses adapted to replicate in other primates to due to ongoing genetic conflict between hosts and pathogens [34]. To ask if human TRIM34 may restrict lentiviruses more broadly, including Simian Immunodeficiency Viruses (SIVs) that are adapted to infect Old World Monkeys, we assayed human TRIM34 for restriction of SIVagm and SIVmac in THP-1 cells. In contrast to human or rhesus TRIM5α, human TRIM34 overexpression inhibits replication of SIVmac signifi-cantly (Fig 3D). Similarly, replication of SIVagm is significantly blocked by human TRIM34 overexpression, although this restriction is not as potent as rhesus TRIM5α restriction of this virus (Fig 3E). Therefore, human TRIM34 retrovirus restriction activity is not limited to HIV-1 CA mutants, as TRIM34 restricts at least two SIVs in addition to the HIV-1 N74D CA mutant. Moreover, because SIVmac is restricted by TRIM34, yet is known to bind CPSF6 [13],

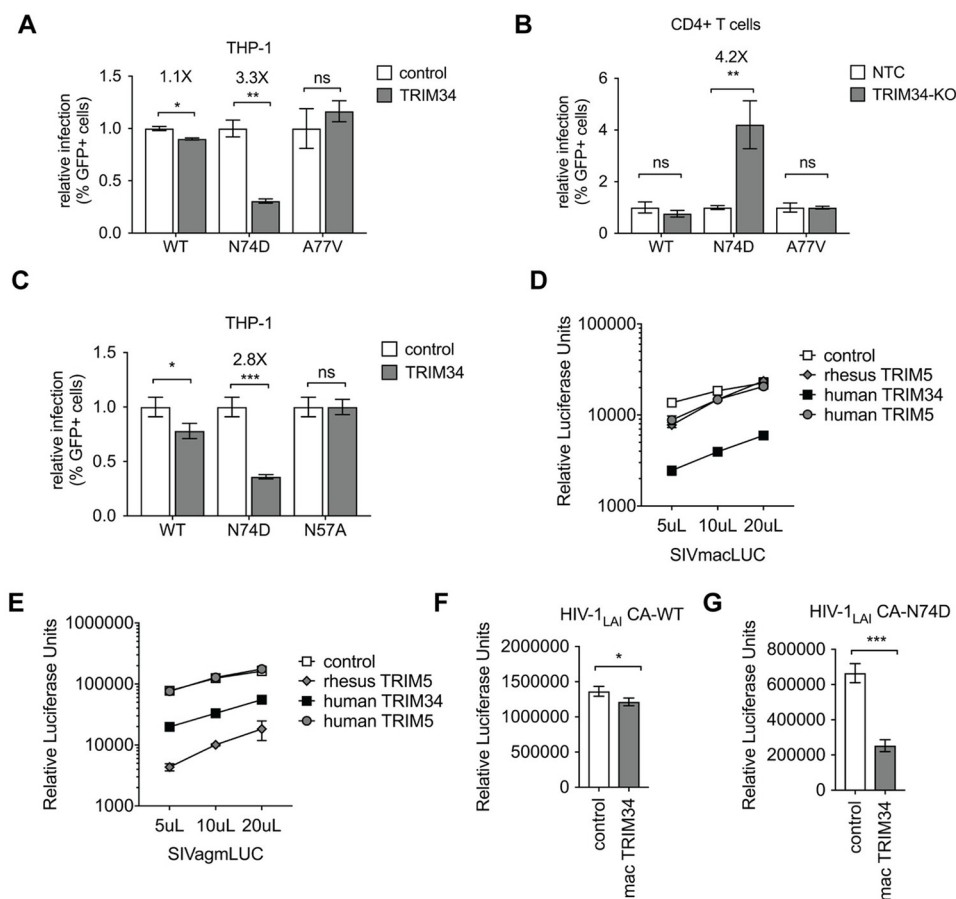

**Fig 3. TRIM34 inhibits a range of primate lentiviruses independent of CPSF6.** A: THP-1 cells stably-overexpressing TRIM34 (gray bars) or control cells (white bars) were infected with WT, N74D or A77V HIV-1 and levels of infection assayed 2 days post-infection by flow cytometry. The relative infection is normalized to the average infection in the control cells for each virus. Data are represented as the mean+/- s.d. from triplicate infections. B: Primary, activated CD4+ T cells were electroporated with Cas9-RNP complexes targeting TRIM34 (gray bars: TRIM34-KO) or Non-Targeting Control crRNAs (white bars: NTC). 2 days later edited CD4+ T cell pools were infected with GFP reporter HIV viruses (WT, N74D or A77V) and infection levels assayed 2 days later by flow cytometry. Data is shown as infection levels relative to the Non-Targeting Control infections for each virus (relative infection). TRIM34-KO pool edited at 52% (ICE KO-score). Data are represented as the mean +/- s.d. from triplicate infections. C: THP-1 cells stably-overexpressing TRIM34 (gray bars) or control cells (white bars) were infected with WT, N74D or N57A HIV-1 and levels of infection assayed 2 days post-infection by flow cytometry. The relative infection is normalized to the average infection in the control cells for each virus. Data are represented as the mean+/- s.d. from triplicate infections. D and E: THP-1 cells were transduced with lentiviral vectors encoding rhesus TRIM5α (gray diamonds), human TRIM5α (gray circles), human TRIM34 (black squares) or a control vector (white squares). Each cell pool was infected with VSV-G pseudotyped SIVmacLUC (D) or SIVagmLUC (E) Luciferase reporter viruses at 3 viral doses as indicated. Levels of infectivity were assayed 2 days later by luciferase assay (RLU = Relative Luciferase Units). Data are represented as the mean +/- s.d. from triplicate infections. F and G: THP-1 cells stably-overexpressing rhesus macaque TRIM34 (mac TRIM34—gray bars) or control cells (white bars) were infected with WT (F) or N74D (G) HIV-1 and levels of infection assayed 2 days post-infection by luciferase assay. Data are represented as the mean+/- s.d. from triplicate infections. $P$ values were determined using two-sided unpaired $t$-tests (ns = not significant, $^*P<0.05$, $^{**}P<0.01$, $^{***}P<0.001$, $^{****}P<0.0001$).

this further supports the model that TRIM34 restriction sensitivity is not necessarily linked to CPSF6 binding of HIV or SIV capsids.

Given the lack of positive selection in TRIM34 in primates [29, 30], one might expect that TRIM34 from non-human primate could substitute for human TRIM34 in restriction of HIV-1

N74D. To test this hypothesis, we tested restriction of HIV viruses in THP-1 cells overexpressing rhesus macaque TRIM34 (mac TRIM34). While there is little effect of mac TRIM34 overexpression on wild type HIV-1 (Fig 3F), we observe significant restriction of the N74D HIV-1 virus (Fig 3G). Therefore, TRIM34 restriction is not specific to humans as TRIM34 from at least one other primate species is also capable of retroviral restriction. We do not see a species-specificity for TRIM34 restriction, consistent with the lack of positive selection signature for this gene.

## TRIM34 restriction requires TRIM5α

TRIM34 is a close paralog of TRIM5α which is known to dimerize and form higher-order oligomers [8]. Furthermore, TRIM34 has been shown to be able to interact with TRIM5α in cells both in a yeast two-hybrid assay [9] as well as in immunoprecipitation studies [10, 29]. As TRIM34 lacks a signal of positive selection in the B30.2 PRY/SPRY domain [29, 30], we hypothesized that TRIM34 might require TRIM5α to restrict HIV-1 infection. This model is particularly intriguing as the PRY/SPRY domain of TRIM34 has a deletion in the v1 loop (see Fig 1D), a region shown to be critical for mediating specificity of capsid recognition by TRIM5α [31]. To test this model, we compared the ability of TRIM34 to restrict the HIV-1 N74D virus in cells both with and without TRIM5α. We introduced a TRIM5α or Control CRISPR/Cas lentiviral vector into THP-1 cells overexpressing TRIM34 (Fig 4A). Infection of these TRIM34-overexpressing, *TRIM5*-KO THP-1 pools demonstrates that the restriction of viral replication measured for the HIV-1 N74D capsid mutant is lost when TRIM5α is missing (Fig 4A). In contrast, we observe little effect of TRIM34 overexpression (1.2-Fold) or *TRIM5*-KO in THP-1 cells infected with the WT virus (Fig 4B), consistent with there being no significant restriction of HIV-1 WT capsid by either TRIM34 or TRIM5α in cells lacking IFN stimulation. Therefore, TRIM34 restriction of the HIV-1 N74D capsid mutant is dependent on TRIM5α.

To ask if this requirement of TRIM5α for TRIM34-mediated restriction is also important in primary cells, we knocked out either TRIM34 or TRIM5 in primary CD4+ cells and infected each cell pool with WT, N74D or P90A in comparison to a control cell pool (Fig 4C; TRIM34-KO 69% ICE KO-score, TRIM5-KO 87% ICE KO-score). Indeed, consistent with a requirement of TRIM5α for the TRIM34-medicated restriction, we find that knockout of either TRIM34 or TRIM5α is sufficient to rescue infection with the N74D capsid mutant (Fig 4C) while neither knockout has a significant effect on WT HIV-1 infection (Fig 4C).

We also examined this question in primary monocyte-derived dendritic cells (MoDCs) by transducing them with shRNA lentiviral vectors, resulting in stable knockdown of TRIM34 (TRIM34 KD) or TRIM5α (TRIM5 KD) (Fig 4D). In addition, to more directly ask if TRIM34 and TRIM5α work together or synergistically, we challenged cells in which both TRIM34 and TRIM5α were depleted simultaneously (TRIM5/TRIM34 DKD). Similar to our results in THP-1 and primary CD4+ T cells, infection with the HIV-1 N74D capsid mutant can be rescued by knockdown of either TRIM34 or TRIM5α in MoDCs (Fig 4D). In contrast, knockdown of either TRIM34 or TRIM5α does not have any effect on WT HIV-1 capsids (Fig 4E). Further, TRIM34 and TRIM5α act in the same pathway, rather than synergistically, to inhibit the HIV-1 N74D capsid mutant since a double-knockdown in MoDCs does not show any additional rescue (Fig 4D).

Finally, we asked if TRIM5α restriction of the CypA-binding deficient P90A capsid mutant virus depends on TRIM34. Consistent with the results of our initial screen with P90A (Fig 1B) and with other recently published results on TRIM5α-sensitivity of this virus [12], we find that P90A is more sensitive to TRIM5α restriction than HIV-1 WT (Fig 4C: CD4+ T Cells; Fig 4F: MoDCs). However, this restriction activity is independent of TRIM34 as TRIM34 knockout or knockdown has little to no effect on P90A (Fig 4C: CD4+ T Cells; Fig 4F: MoDCs). These data

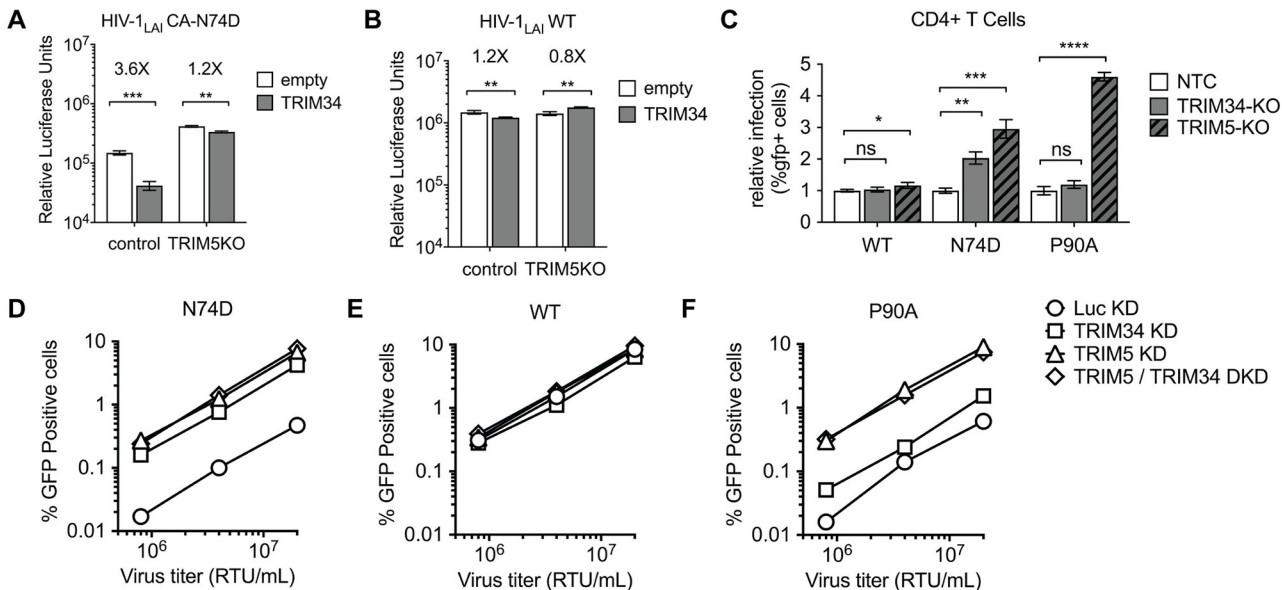

**Fig 4. TRIM34 requires TRIM5α to restrict the N74D virus.** A and B: THP-1 cells stably-overexpressing TRIM34 (gray bars) or control cells (white bars) were transduced with sgRNA-encoding lentiviral vectors targeting TRIM5α (TRIM5KO) or a control. *TRIM5* alleles were edited at 70% for both control and TRIM34-overexpressing cells as determined by ICE analysis. Cell pools were infected with the N74D-LUC virus (A) or WT-LUC virus (B) and levels of infection assayed 2 days post-infection by luciferase assay (Relative Luciferase Units). Data are represented as the mean +/- s.d. from triplicate infections. C: Primary, activated CD4+ T cells were electroporated with Cas9-RNP complexes targeting TRIM34 (gray bars), TRIM5α (hatched, dark gray bars) or NTC crRNAs (white bars). 2 days later edited CD4+ T cell pools were infected with GFP reporter HIV-1 viruses (WT, N74D or P90A) and infection levels assayed 2 days later by flow cytometry. TRIM34-KO pools were edited at 69% and TRIM5-KO pools were edited at 87% (ICE KO-Scores). The relative infection is normalized to the average infection in the control cells for each virus. Data are represented as the mean +/- s.d. from triplicate infections. D, E and F: Monocyte-derived dendritic cells were simultaneously transduced with a lentiviral vector encoding shRNAs targeting TRIM34 or luciferase control (Luc) and the other shRNA vector specific for TRIM5α or Luc, for the knockdown as indicated. The pooled cells were challenged with VSV-G pseudotyped HIV-1 vectors expressing GFP reporter and containing CA-N74D (D), WT CA (E), or CA-P90A (F), across a range of viral inputs normalized for RT activity (RTU: Reverse Transcriptase Units/mL). The percentage of GFP-positive cells was determined 2 days later by flow cytometry. *P* values were determined using two-sided unpaired *t*-tests (ns = not significant, *$P<0.05$, **$P<0.01$, ***$P<0.001$, ****$P<0.0001$).

suggest that TRIM34 restriction depends on TRIM5α, but that TRIM5α restriction does not depend on TRIM34. Therefore, there is asymmetry in the TRIM5α/TRIM34 relationship as their interdependence is not equivalent across restriction activities.

## TRIM34 and TRIM5α complexes colocalize with N74D capsids

Since TRIM34 restriction depends on TRIM5α (Fig 4), we tested the hypothesis that TRIM34 and TRIM5α colocalize with each other and with incoming HIV-1 capsids during infection. First, we created HeLa cell lines stably-expressing YFP-TRIM5α and HA-TRIM34 to ask if both proteins localize to the same subcellular compartment in cells. We find that TRIM34 localizes to cytoplasmic puncta, commonly referred to as cytoplasmic bodies, both together with and separate from TRIM5α (Fig 5A–"Mock"; white triangles are puncta with colocalization of TRIM34 and TRIM5α).

Given that TRIM34, together with TRIM5α, restricts N74D capsids but not WT capsids, we hypothesized that this differential restriction could be due to the localization of TRIM34 and TRIM5α to N74D capsids that does not occur, or does not occur to the same extent, as with WT capsids. To test this hypothesis, HA-TRIM34 and YFP-TRIM5α stably-expressing HeLa cells were infected with WT or the N74D viruses and 2 hours later colocalization of both TRIM34 and TRIM5α with each capsid was measured. We observe colocalization of TRIM34

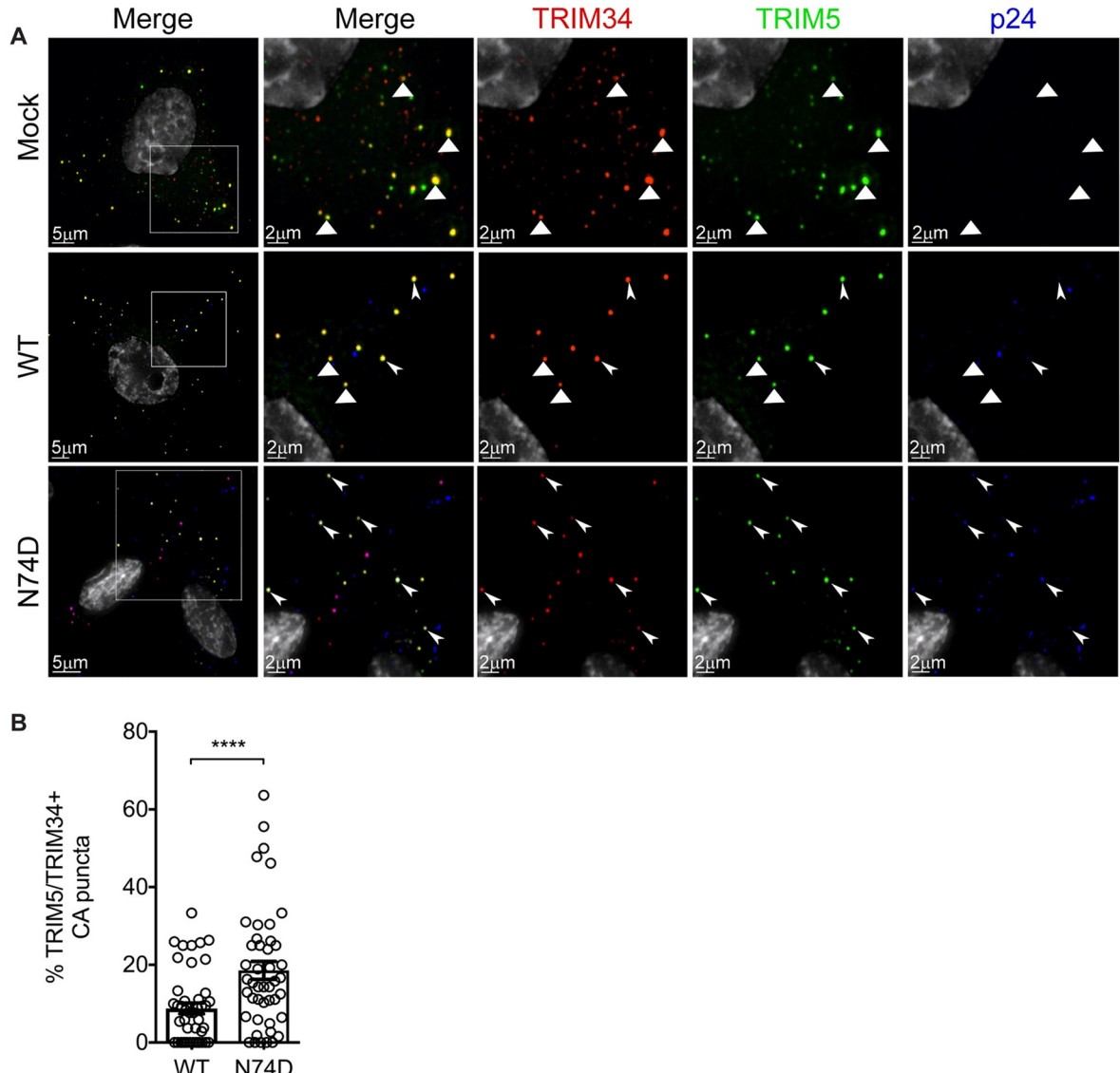

**Fig 5. TRIM34 colocalizes more frequently with the restricted HIV-1 N74D capsid.** HeLa cells were transduced to express YFP-TRIM5α (green) and HA-TRIM34 stably. They were plated on coverslips and synchronously infected with VSV-G pseudotyped HIV-1 with WT or N74D viruses as indicated. At 2 hpi, cells were fixed and stained for viral capsid protein p24$^{gag}$+ (blue) and HA-TRIM34 tag (red). Images from 15–20 cells were collected per condition in three independent biological replicates. A: Representative images for mock-infected cells (top row), WT-infected cells (middle row), and N74D-infected cells (bottom row). White triangles indicate colocalization of TRIM34 and TRIM5α in the zoomed in images for each channel. Triple colocalization between TRIM34, TRIM5α and p24$^{gag}$+ are indicated by arrowheads in the zoomed in images for each channel. B: Quantification of percent p24$^{gag}$+ colocalizing with both TRIM34 and TRIM5α for the WT and N74D virus. Error bars represent s.e.m. of all events collected across three biological replicates for each condition. *P* value was determined by a two-sided unpaired *t*-test.

and TRIM5α together with both WT and N74D HIV-1 capsids in the cell cytoplasm (Fig 5A– WT and N74D; white arrowheads are puncta with p24$^{gag}$, TRIM34 and TRIM5α). Quantification of the number of p24$^{gag}$+ puncta that are also positive for TRIM34 and TRIM5α shows that colocalization of both TRIM proteins with p24$^{gag}$ occurred more frequently for N74D capsids than for WT capsids (Fig 5B). Therefore, TRIM34 and TRIM5α are present together with incoming N74D HIV-1 capsids in the cytoplasm of infected cells.

## Discussion

We identified TRIM34, a *TRIM5* paralog, as an HIV-1 restriction factor capable of inhibiting infection by an HIV-1 N74D capsid mutant virus as well as several lentiviruses from monkeys. This block occurs before the completion of reverse transcription and is a constitutive block to infection as the restriction is observed in both IFN-stimulated and unstimulated cells. More-over, TRIM34 restriction also occurs in primary, activated CD4+ T cells and monocyte-derived dendritic cells (MoDCs). The antiviral activity of TRIM34 is independent of the ability of viral capsids to bind CPSF6 but the antiviral restriction activity of TRIM34 requires TRIM5α in all cells tested. Finally, TRIM34 and TRIM5α colocalize in cells and preferentially localize to restricted N74D capsids as compared to WT capsids.

### TRIM34 is a capsid-targeting HIV-1 restriction factor

While we did not identify here the factor responsible for enhanced sensitivity of the N74D mutant to IFN, we do uncover IFN-independent restriction by a novel capsid-targeting restriction factor, TRIM34. *TRIM34* is a close paralog of the well-studied capsid-targeting restriction factor *TRIM5*. Previously, modest restriction of SIVmac and the non-primate lentivirus, Equine Infectious Anemia Virus (EIAV), by human TRIM34 was described but no effect of TRIM34 was observed on either WT HIV-1 or a G89V capsid mutant virus [9]. Our data suggest that TRIM34, together with TRIM5α, mediates all or most of the block to replication of the N74D capsid mutant observed in primary cells, including CD4+ T cells and MDMs, first described by Ambrose et al. [18]. The HIV-1 N74D capsid mutation is partially rescued by the capsid-binding inhibitor PF74 [35, 36], which suggests the possibility that PF74 might compete with TRIM34 restriction of this capsid. However, since the N74D mutation has pleiotropic effects, this rescue of the N74D mutant could also be independent of TRIM34.

TRIM5α knockout or knockdown rescues the HIV-1 P90A capsid mutant in human primary cells suggesting that at least one function of CypA binding is evasion of TRIM5α-mediated restriction [12]. Our screen with P90A did not uncover any additional factors, other than TRIM5α, suggesting that the increased IFN sensitivity of this mutant is entirely due to TRIM5α restriction.

### TRIM34 as a species barrier

The site that sensitizes HIV-1 capsid to TRIM34 restriction, N74, is highly-conserved across HIV-1 and SIV strains [13]. Mutations at this site may sensitize capsids to TRIM34/TRIM5α restriction and therefore have been selected against. However, this effect is independent of CPSF6 binding to capsids as loss of binding to CPSF6 is not sufficient for restriction by TRIM34 (see the A77V mutant in Fig 3A and 3B and the N57A mutant in Fig 3C). Our results are consistent with the finding that WT capsids do not become IFN-hypersensitive in CPSF6 knockout cells [19]. Therefore, our data support a model in which CPSF6 binding plays a role in integration targeting [15–17, 37, 38] but it does not shield HIV-1 capsids from capsid-targeting restrictions. In contrast, loss of CypA binding sensitizes HIV-1 capsids to TRIM5α restriction [12]. However, loss of CypA binding is not sufficient for TRIM5α restriction of all primate lentiviruses as SIVmac is not restricted by TRIM5α even though SIVmac capsids do not bind CypA to any appreciable affinity [17]. These data together highlight the complexity of capsid adaptation to host restrictions where multiple, independent pathways of adaptation to multiple restrictions may occur across divergent strains.

The capsid-targeting restriction factors MxB and TRIM5α together with other restriction factors likely constrain HIV-1 evolution *in vivo* and provide a significant host barrier that HIV-1 must adapt to in order to successfully establish infection during transmission to a new

host [39]. Capsid is a dominant determinant of species tropism for cross-species replication of primate lentiviruses such as SIVmac [40]. Our results suggest that TRIM34, like TRIM5α, is a lentiviral restriction factor that HIV-1 has adapted to in order to replicate efficiently in human cells since SIVs are also sensitive to human TRIM34. Further, TRIM34 may act more broadly across primates to restrict lentiviral infection, as rhesus macaque TRIM34 similarly blocks N74D replication (Fig 3G). Of note, rhesus macaque TRIM34 shows a similar pattern of restriction as human TRIM34 thereby suggesting that perhaps the specificity of TRIM34 restriction is determined by its interaction with TRIM5α. However, overexpression of TRIM5α alone is not sufficient to mediate restriction (Fig 3D and 3E). Taken together our results suggest that primate TRIM34, perhaps together with TRIM5α, could represent a significant barrier to cross-species transmission of primate lentiviruses. This interpretation is supported by the observation that N74 is a highly-conserved residue in lentiviral capsids.

## TRIM34 restriction requires TRIM5α

Of particular interest, we find that TRIM34 restriction depends on TRIM5α. More broadly, TRIM proteins are a large gene family [41] and potentially-important functional interactions between family members likely have been overlooked. Our data show that TRIM34 can restrict lentiviral capsids but that this activity requires expression of TRIM5α, a close paralog of TRIM34. The ability of TRIM family members to hetero-oligomerize has been known for some time, but our screen highlights the power of an unbiased approach such as HIV-CRISPR screening to uncover previously-unappreciated functional interactions between TRIM family members.

This dependence of TRIM34 on TRIM5α raises several key questions including: 1) how does TRIM34 allow for specific restriction of the N74D capsid mutant viruses? and 2) why does this restriction depend on TRIM5α? We propose two potential models that are consistent with our data that are not mutually exclusive. First, it is possible that TRIM34 changes the specificity of TRIM5α such that it can now recognize the N74D capsid mutant virus better and/or more efficiently. TRIM34 can bind to HIV-1 capsids through its SPRY domain [29, 42]. However, when compared to TRIM5α, human TRIM34 has a deletion in the v1 loop of the SPRY domain that is important for determining capsid binding specificity. Further, TRIM34 is not itself evolving under positive selection, an evolutionary signature that would be consistent with a direct protein interface with a viral pathogen. Instead, it may be that the TRIM34 SPRY does not itself make significant contact with the N74D capsid but that TRIM34 complexing with TRIM5α may affect recognition of this capsid specifically by human TRIM5α.

A second possibility is that TRIM34 stabilizes TRIM5α, thereby allowing it to restrict N74D capsids. This model does not require any change in specificity of binding to capsid, *per se*, but rather results from changes in kinetics and/or entry of HIV-1 capsid mutants. Nuclear entry by HIV is highly dependent on cell type, including nuclear pore composition, and capsid sequence [22]. In this model, WT HIV-1 capsids are not sensitive to TRIM34 restriction as they transit the cytoplasm more efficiently and/or via a different nuclear entry route that allows escape from restriction. As proposed by Sultana et al., an accelerated rate of uncoating could be a mechanism of escape from cellular restrictions, particularly those that target the HIV-1 capsid [43]. In support of this model, it has been shown that while human TRIM5α has a short half-life and turns over quickly in cells, TRIM34 is significantly more stable [29]. It may be that TRIM34 stabilizes TRIM5α, thereby allowing it to restrict HIV capsids that transit differently than WT into the nucleus. In support of this model, slower uncoating kinetics of the N74D capsid mutant have been observed [44]. However, conflicting data shows that the N74D

capsid mutant has been reported to have the same intrinsic stability as the WT capsid as measured by an *in vitro* uncoating assay [45].

A further implication of the interaction of TRIM34 and TRIM5α is that TRIM34 may play a role in sensing infection, the pattern recognition function of TRIM5α. The N74D capsid mutant has been proposed to trigger sensing pathways leading to IFN and other cytokine production in macrophages [20] although this phenotype of sensing of the N74D mutant occurs in macrophages has been challenged [33]. Work is currently underway to understand the potential role of TRIM34 in HIV sensing and pattern recognition/innate immune activation more broadly.

## Materials and methods

### Cells and cell culture

All cells were incubated in humidified, 5% $CO_2$ incubators at 37 ˚C. The THP-1 monocytic cell line (ATCC) was cultured in RPMI (Invitrogen) with 10% FBS, Pen/Strep, 10 mM HEPES, 0.11 g/L sodium pyruvate, 4.5 g/L D-Glucose and Glutamax. 293T (ATCC CRL-3216) and TZM-bl cells (ATCC 8129) were cultured in DMEM (Invitrogen) with 10% FBS and Pen/Strep. Puromycin selections in THP-1 cells were done at 0.5–1 ug/mL. The identity of THP-1 cells was confirmed by STR profiling (Fred Hutch Research Cell Bank). For the MoDC-related work, HEK293 cells (American Type Culture Collection) were cultured in DMEM supplemented with 10% heat-inactivated FBS, 20 mM GlutaMAX-I, 1 mM sodium pyruvate, 1× MEM non-essential amino acids and 25 mM HEPES, pH 7.2. HeLa and 293T cell lines utilized in the immunofluorescence assays were obtained from the American Type Culture Collection and were cultured in Dulbecco's modified Eagle's medium (DMEM) supplemented with 10% fetal bovine serum (FBS) (Atlanta Biologicals), 100 U/ml penicillin, 100 µg/ml streptomycin, and 10 µg/ml ciprofloxacin. To generate MoDCs, Peripheral Blood Mononuclear Cells (PBMCs) were isolated from leukopaks by gradient centrifugation on Lymphoprep (Axis-Shield Poc AS #AXS-1114546). CD14+ PBMCs were enriched using anti-CD14 antibody microbeads (Miltenyi Biotec #130-050-201), according to manufacturer's protocol. The enriched CD14+ cells were cultured at a density of $2 \times 10^6$ cells/mL, in RPMI-1640, supplemented with 5% heat-inactivated human AB+ serum (Omega Scientific), 20 mM GlutaMAX-I, 1 mM sodium pyruvate, 1× MEM non-essential amino acids and 25 mM HEPES pH 7.2. The addition of 1:100 cytokine-conditioned media containing hGM-CSF and hIL-4 to the culture promoted the differentiation of CD14+ cells into MoDCs. These cytokine-conditioned media were produced from HEK293 cells stably transduced with pAIP-hGMCSFco (Addgene no. 74168) or pAIP-hIL4-co (Addgene no. 74169), as previously described [46, 47]. For the analysis of reverse transcription products, a clonal TRIM34-KO THP-1 cell line was created through single-cell sorting of Cas9/RNP electroporated pools into 96-well plates to create individual clonal lines (BD FACS Aria II–Fred Hutch Flow Cytometry Core). A clonal KO line was identified through ICE Editing Analysis (Synthego). Universal Type I Interferon Alpha was obtained from PBL Assay Science (Catalog No. 11200–2), diluted to $10^5$ Units/mL in sterile-filtered PBS/1% BSA according to the activity reported by manufacturer and frozen in aliquots at −80˚C.

### Human blood

For CD4+ T cell experiments, whole blood from anonymous donors was obtained from Blood-Works Northwest, total PBMCs were isolated using the density gradient centrifugation method with Histopaque-1077 (Sigma-Aldrich #10771) and CD4+ T cells were isolated using EasySep Human CD4+ T cell isolation kit (StemCell Technologies #17952). Cells were resuspended to $2.5 \times 10^6$ cells/mL in RPMI complete media supplemented with 10% FBS, Glutamax

and Pen/Strep and with 100 U/mL recombinant human IL-2 (Roche; Sigma # 10799068001). For monocyte-derived dendritic cell (MoDC), leukopaks were acquired from anonymous, healthy blood donors (New York Biologics).

## Ethics statement

All primary cell data is from anonymous blood donors and is classified as "human subjects exempt" research by the University of Massachusetts Medical School or Fred Hutchinson Cancer Research Center Institutional Review Boards, according to National Institutes of Health (NIH) guidelines (http://grants.nih.gov/grants/policy/hs/faqs_aps_definitions.htm). No animal work was done.

## Plasmids

HIV infectious clones based on the LAI strain of HIV-1 (pLAI) were used in this study. The pLAI GFP3 backbone encodes the green fluorescent protein (GFP) gene in place of the *nef* gene [48]. The pLAI GFP3* WT, pLAI GFP3* N74D, pLAI GFP3* A77V and pLAI GFP3* N57A proviruses were provided by Masahiro Yamashita and are described in [33]. The P90A CA mutation was introduced into pLAI GFP3 using standard cloning procedures as described previously [49]. The luciferase envelope-defective reporter proviral N74D plasmid was cloned from pLAI GFP3* N74D by BssHI and SalI digest and cloned into BruLuc2deltaEnv [48]. The SIVmacLUC E-R- and SIVagmLUC E-R- plasmids were a gift from Ned Landau [50]. The lentiCRISPRv2 plasmid was a gift from Feng Zhang (Addgene #52961). pMD2.G and psPAX2 were gifts from Didier Trono (Addgene #12259/12260). lentiCRISPRv2 constructs targeting genes of interest were cloned into BsmBI-digested lentiCRISPRv2 by annealing complementary oligos with overhangs that allow directional cloning into lentiCRISPRv2. TRIM34 oligos used were: TRIM34KO_1: TRIM34_1 Sense CACCGGTCAAGTTGAGCCCAGACAA and TRIM34_1 Antisense AAACTTGTCTGGGCTCAACTTGACC; TRIM34KO_2: TRIM34_2 Sense CACCGGAGTAACTGATACCACACAC and TRIM34_2 Antisense AAACGTGTGTG GTATCAGTTACTCC). TRIM5 oligos used were: TRIM5KO: TRIM5 Sense CACCGGTTGA TCATTGTGCACGCCA and TRIM5 Antisense AAACTGGCGTGCACAATGATCAACC). The lentiviral pHIV-dTomato (Addgene #21374) expression vector was a gift from Bryan Welm (Addgene plasmid #21374; http://n2t.net/addgene:21374; RRID:Addgene_21374). The human TRIM34 cDNA was purchased from Genscript (NM_001003827.1). Human TRIM34 was cloned into pHIV/dTomato using NotI and XmaI sites with an HA tag encoded at the N-terminus. The macaque TRIM34 cDNA (NM_001205182.3) was synthesized by Integrated DNA Technologies (IDT) and cloned into pHIV/dTomato using NotI and XmaI sites. For HeLa immunofluorescence assays, YFP- TRIM5α and HA-TRIM34 were cloned in frame, into retroviral vectors EXN and YXN as described previously [51].

## Virus and lentivirus production

Replication-competent HIV-1 viruses were produced as previously described [6]. Briefly, 293T cells (ATCC) were plated at $2 \times 10^5$ cells/mL in 2 mL in 6-well plates one day prior to transfection using TransIT-LT1 reagent (Mirus Bio LLC) with 3 μL of transfection reagent per μg of DNA. For HIV-1 production, 293Ts were transfected with 1 ug/well proviral DNA. One day post-transfection media was replaced. Two- or three- days post-transfection viral supernatants were clarified by centrifugation (1000 g) and filtered through a 20 μm filter. For Benzonase-treated viral preps, viral supernatants were incubated with 1 uL Benzonase (Sigma Aldrich #E1014) per 1mL of viral supernatant for 30 minutes at 37˚C after dilution in 10X Benzonase Buffer (500mM Tris-HCl pH 8.0, 10mM MgCl$_2$, 1 mg/mL Bovine Serum Albumin). For HIV-1

vectors used in Fig 4D–4F, HEK293 cells were seeded at 75% confluency in 6-well plates. Transfections were performed with 6.25 μL TransIT LT1 transfection reagent (Mirus) in 250 μL Opti-MEM (Gibco) with 2.49 μg total plasmid DNA. 2.18 μg of env-defective HIV-1 provirus containing GFP reporter was cotransfected with 0.31 μg pMD2.G VSV G plasmid (Addgene #12259). Simian immunodeficiency virus (SIV)−VLPs containing Vpx were produced by the transfection of 2.18 μg pSIV−Δpsi/94env/ΔVif/ΔVpr (Addgene #132928) and 0.31 μg pMD2.G plasmid. 16 hours post-transfection, the culture media was changed to the media for MoDC culture. Viral supernatant was harvested 2 days later, filtered through a 0.45 μm filter and stored at −80 ˚C. For lentiviral preps (lentiCRISPRv2 and pHIV), 293Ts were transfected with 667 ng lentiviral plasmid, 500 ng psPAX2 and 333 ng MD2G. For PIKA-HIV library preps, supernatants from $20 \times 6$ well plates were combined and concentrated by ultracentrifugation. 30 mL of supernatant per SW-28 tube were underlaid with sterile-filtered 20% sucrose (1 mM EDTA, 20 mM HEPES, 100 mM NaCl, 20% sucrose) and spun in an SW28 rotor at 23,000 rpm for 1 hr at 4˚C in a Beckman Coulter Optima L-90K Ultracentrifuge. Supernatants were decanted, pellets resuspended in DMEM over several hours at 4˚C and aliquots frozen at −80˚C. All viral and lentiviral infections and transductions, except those in Figs 4D–4F or 5, were done in the presence of 20 μg/mL DEAE-Dextran (Sigma #D9885).

## HIV-CRISPR screening & screen analysis

HIV-CRISPR Screening and Analysis was performed as described [6] with the ISG-enriched PIKA-HIV library with the exception that the viral dose used in each screen allowed for infection of only ~10–30% of cells for each capsid mutant virus. All screens were performed in a clonal THP-1 ZAP-KO cell line [6]. Analysis of screen data was performed as previously described [6] with the exception that single mismatches were allowed when assigning reads to each sample during multiplexing. Sequencing data is available through the Gene Expression Omnibus (GEO−GSE140467).

## Transduction with lentiviral knockdown, knockout and overexpression vectors

For stable overexpression of TRIM34, THP-1 cells were transduced with pHIV/dTomato-TRIM34 or pHIV/dTomato empty vector lentiviral preps. 2–5 days post-transduction cells were sorted for high dTomato expression to select for high-expressing populations. Transduced cells were resorted as needed. For shRNA knockdown in MoDCs, $2 \times 10^6$ CD14$^+$ monocytes/mL were transduced with a 1:4 volume of SIV−VLPs and a 1:4 volume of knockdown lentivectors, as indicated. The SIV−VLPs were added to transfer Vpx to the cells in order to overcome restriction by SAMHD1 against lentiviral transduction [52, 53]. Transduced cells were then selected with both 3 μg/mL puromycin (InvivoGen #ant-pr-1) and 10 μg/mL blasticidin (InvivoGen #ant-bl-1) for 3 days, starting at day 3 post-transduction. To generate stable HeLa cell lines used in immunofluorescence assays, a retrovirus was prepared by transfecting equal amounts of VSV-G, pCigB packaging plasmid, EXN HA-TRIM34 or YXN YFP-TRIM5 into HEK293T cells. Viral supernatant was harvested and filtered through 0.45 μm filters (Milipore) and applied to HeLa cells. 48 hrs after transduction, G418 was added to the cells, and following selection, cells were collected to check protein expression by Western blotting. To generate KO pools, THP-1 cells were transduced with lentiCRISPRv2 vectors and selection in Puromycin. KO cell pools were validated using genomic editing analysis as described below (Editing Analysis).

## Cas9/RNP electroporation

Multiplexed Gene Knockout Kits targeting TRIM34 and TRIM5 were purchased from Synthego. The TRIM5 Kit includes the following sgRNA sequences: AAUCUUGCUUAACG UACAAG, UGGCCACAGUCUAGACUCAA and GAGGCAGUGACCAGCAUGGG. Primers used to amplify the genomic locus and sequencing for TRIM5 were: GAAAAGCCCTTAT TACCAGG (For) and GAGAATCCATGACTTGGAAG (Rev). The TRIM34 Kit includes the following sgRNA target sequences: AGGUCUUGUGGUUUGCAGUG, AGGGGUUAAU GUAAAGGAGG and GGGAACUGAUCCGGCACACA. TRIM34 amplification and sequencing primers were provided with the kit. CD4+ T cells were activated for 3 days with 10 ug/mL of plate-bound anti-CD3 (Tonbo Biosciences, clone UCHT1; #70-0038-U100) and 5 ug/mL of diffused anti-CD28 (Tonbo Biosciences, clone CD28.2; #70-0289-U100). For each electroporation 1 x $10^6$ CD4+ T cells were pelleted by centrifugation at 100 x $g$ for 10 mins and washed once in PBS. Cells were resuspended in 25 uL of CRISPR/Cas9 crRNP complexes that were pre-assembled in P3 Primary Cell Nucleofector Solution (Lonza #V4SP-3096) before electroporation in a single well of a 96-well Nucleocuvette Plate using program EH-115. Each electroporation was diluted with 80 uL of RPMI complete + 125 U/mL IL-2 and allowed to recover for 1–2 hours at 37˚C. Cells resuspended at 2.5 x $10^6$ cells/mL were transferred to a 96-well plate in RPMI complete + 100 U/mL IL-2 and re-activated with anti-CD2/CD3/CD28 beads at a 1:1 ratio (T Cell Activation/Expansion Kit, Milltenyi Biotec #130-091-441). Two days post-electroporation, each well was supplemented with 100 uL of RPMI complete + 100 U/mL IL-2. Beginning from 4 days post-electroporation, cells were maintained and propagated at 1 x $10^6$ cells/mL with fresh RPMI complete + 100 U/mL IL-2 being added every 2–3 days until infection and editing analysis.

## Editing analysis

Cell populations were analyzed for allele editing frequency as previously described [6]. Briefly, genomic DNA was isolated with a QIAamp DNA Mini Kit (Qiagen #51185), amplified by primers surrounding the editing site and sanger sequenced. Editing levels were analyzed by ICE Analysis (Synthego) to obtain an ICE KO-Score (ice.synthego.com/#/).

## Exogenous reverse transcriptase assay

A 5 μL transfection supernatant containing virions was lysed in 5 μL 0.25% Triton X-100, 50 mM KCl, 100 mM Tris–HCl pH 7.4 and 0.4 U/μL RiboLock RNase inhibitor. This viral lysate was then diluted 1:100 in 5 mM $(NH_4)_2SO_4$, 20 mM KCl and 20 mM Tris–HCl pH 8.3. 10 μL of this was then added to a single step, RT−PCR assay with 35 nM bacteriophage MS2 RNA (Integrated DNA Technologies) as a template, 500 nM of each primer (5'-TCCTGCTCAACT TCCTGTCGAG-3' and 5'-CACAGGTCAAACCTCCTAGGAATG-3') and 0.1 μL hot-start Taq DNA polymerase (Promega) in 20 mM Tris–HCl pH 8.3, 5 mM $(NH_4)_2SO_4$, 20 mM KCl, 5 mM $MgCl_2$, 0.1 mg/mL BSA, 1/20,000 SYBR Green I (Invitrogen) and 200 μM dNTPs in a total reaction volume of 20 μL. The RT−PCR reaction was carried out in a Bio-Rad CFX96 cycler with the following parameters: 42 ˚C for 20 min, 95 ˚C for 2 min and 40 cycles (95 ˚C for 5 s, 60 ˚C for 5 s, 72 ˚C for 15 s and acquisition at 80 ˚C for 5 s).

## Viral Infectivity assays

Cells were pre-stimulated with IFNα 24 hours prior to infection where indicated. Virus and 20 μg/mL DEAE-Dextran in RPMI were added to cells, spinoculated for 20 min at 1100x$g$, and incubated overnight at 37˚C. Cells were washed the next day and re-suspended in RPMI

supplemented with IFNα. For infectivity assays using single-cycle viruses in MoDCs, $2.5 \times 10^5$ cells were plated per well, in a 48-well plate, on the day of virus challenge. Media containing VSV G-pseudotyped HIV-1 vectors encoding GFP reporter was added to challenge cells in a total volume of 250 μL per condition. Cells were harvested for flow cytometric analysis by scraping at 48 hours post-challenge with the HIV-1 vectors and fixed in a 1:4 dilution of BD Cytofix Fixation Buffer with phosphate-buffered saline (PBS) without $Ca2^+$ and $Mg2^+$, supplemented with 2% FBS and 0.1% NaN3.

## Flow cytometry

For intracellular p24$^{gag}$ staining, cells were harvested and fixed in 4% paraformaldehyde for 10 min and diluted to 1% in PBS. Cells were permeabilized in 0.5% Triton-X for 10 min and stained with 1:300 KC57-FITC (Beckman Coulter 6604665; RRID: AB_1575987). Data were collected on Accuri C6 (BD Biosciences–U Mass) or a BD FACSCANTO II (Fred Hutch Flow Cytometry Core) and analyzed with FlowJo software. For cell surface marker staining, cells were washed twice in PBS, stained in PBS/1% BSA, incubated at 4°C for 1 hr, washed twice in PBS, and analyzed on the Canto two flow cytometer (Fred Hutch Flow Cytometry Core).

## Luciferase assay

For analysis of Luciferase activity, infected cells were lysed in 100 μL BrightGlo Luciferase reagent (Promega #E2610) and read on a LUMIstar Omega Luminometer.

## qPCR assay for HIV late reverse transcription products

For qPCR analysis of HIV-1 late RT products, THP-1 cells were infected in 6-well plates with Benzonase-treated viral preparations (see Virus production methods). Total DNA was extracted from infected cells approx. 16 hours post-infection with a QIAprep Spin Miniprep kit (Qiagen #27106). HIV-1 cDNA was amplified using TaqMan Gene expression Master Mix (AppliedBiosystems #4369016) with 900nM of each primer: J1 FWD (Late RT F)–ACAAGCT AGTACCAGTTGAGCCAGATAAG, J2 REV (Late RT R)–GCCGTGCGCGCTTCAGCAA GC and 250nM LRT-P (late RT Probe)–FAM-CAGTGGCGCCCGAACAGGGA-TAMRA. qPCR data was collected on an ABI QuantStudio5 Real Time (qPCR) System Instrument.

## Immunofluorescence microscopy

HeLa cells were allowed to adhere to glass coverslips placed in wells of a 24-well plate. Synchronized infection was performed by spinoculation of reporter virus onto cells at 13°C for 2 hours at $1,200 \times g$, after which virus-containing medium was removed and replaced with warm media. Coverslips containing cells were incubated in 37°C for 2 hours, and were subsequently fixed with 3.7% formaldehyde (Polysciences) in 0.1 M PIPES [piperazine-N,N'-bis(2-ethane-sulfonic acid)], pH 6.8. Cells were permeabilized with 0.1% saponin, 10% normal donkey serum, 0.01% sodium azide in PBS. The following primary antibodies were used for immuno-fluorescence: mouse anti-HIV-1 p24$^{gag}$ (Santa Cruz, Cat. # sc-69728), rabbit anti-HA (Sigma, Cat. # H6908). Primary antibodies were labeled with fluorophore-conjugated donkey anti-mouse or anti-rabbit antibody (Jackson ImmunoResearch Laboratories, Inc., West Grove, PA, USA). Images were collected with a DeltaVision microscope (Applied Precision, Issaquah, WA, USA) equipped with a digital camera (CoolSNAP HQ; Photometrics, Tucson, AZ, USA), using a 1.4-numerical aperture (NA) 100x objective lens, and were deconvolved with Soft-WoRx software (Applied Precision, Issaquah, WA, USA).

For each condition in each experiment, 15 Z-stack images were acquired using identical acquisition parameters. Deconvolved images were analyzed using Imaris Software (Bitplane). For analysis of colocalization between p24$^{gag}$ and HA-TRIM34 and/or YFP- TRIM5α, an algorithm was designed to create a three-dimensional surface around p24$^{gag}$ signal. The algorithm was applied to all the images within a given experiment. Colocalization was defined as the presence of a signal intensity above a threshold value, and the same colocalization threshold was maintained for a given channel for all the images and conditions in a particular experiment. Both graphing and statistics calculations were performed in Prism (Graphpad Software, Inc).

## Supporting information

**S1 Table. MAGeCK Gene Scores for all HIV-CRISPR screens.** WT +IFN (Tab 1), P90A +IFN (Tab 2), N74D +IFN (Tab 3), N74D noIFN (Tab 4). *NegLog10PosGeneScore* column shows -log$_{10}$MAGeCK Gene Scores.
(XLSX)

## Acknowledgments

We thank Masa Yamashita, ADARC, for providing capsid mutant proviral plasmids, including the unpublished HIV-1 P90A capsid mutant provirus, Daryl Humes for help with CD4 T cell preparations and electroporations, Sydney Fine for performing editing analysis on edited T cell pools and the Fred Hutch Genomics Core, including Jeff Delrow and Ryan Basom, for help with deep sequencing and HIV-CRISPR screen analysis. We are also grateful to the anonymous blood donors who contributed cells to this study.

## Author Contributions

**Conceptualization:** Molly Ohainle, Kyusik Kim, Sevnur Komurlu Keceli, Abby Felton, Ed Campbell, Jeremy Luban, Michael Emerman.

**Data curation:** Molly Ohainle, Kyusik Kim, Sevnur Komurlu Keceli, Abby Felton.

**Formal analysis:** Molly Ohainle, Kyusik Kim, Sevnur Komurlu Keceli.

**Funding acquisition:** Molly Ohainle, Ed Campbell, Jeremy Luban, Michael Emerman.

**Investigation:** Molly Ohainle, Kyusik Kim, Sevnur Komurlu Keceli, Abby Felton.

**Methodology:** Molly Ohainle, Kyusik Kim, Sevnur Komurlu Keceli, Ed Campbell, Jeremy Luban, Michael Emerman.

**Project administration:** Molly Ohainle, Ed Campbell, Jeremy Luban, Michael Emerman.

**Resources:** Molly Ohainle, Kyusik Kim, Sevnur Komurlu Keceli, Abby Felton, Ed Campbell, Jeremy Luban, Michael Emerman.

**Software:** Molly Ohainle, Sevnur Komurlu Keceli.

**Supervision:** Molly Ohainle, Ed Campbell, Jeremy Luban, Michael Emerman.

**Validation:** Molly Ohainle, Kyusik Kim, Sevnur Komurlu Keceli, Abby Felton, Ed Campbell, Jeremy Luban, Michael Emerman.

**Visualization:** Molly Ohainle, Kyusik Kim, Sevnur Komurlu Keceli.

**Writing – original draft:** Molly Ohainle, Michael Emerman.

**Writing – review & editing:** Molly Ohainle, Kyusik Kim, Sevnur Komurlu Keceli, Ed Campbell, Jeremy Luban, Michael Emerman.

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
