## [Decision Letter · Decision Letter 0]

14 Jan 2020

Dear Dr. Ohainle,

Thank you very much for submitting your manuscript "TRIM34 acts with TRIM5α to restrict HIV-1 and SIV capsids" (PPATHOGENS-D-19-02127) for review by PLOS Pathogens. Your manuscript was fully evaluated at the editorial level and by independent peer reviewers. The reviewers appreciated the attention to an important problem, but raised some substantial concerns about the manuscript as it currently stands. These issues must be addressed before we would be willing to consider a revised version of your study. We cannot, of course, promise publication at that time.

We therefore ask you to modify the manuscript according to the review recommendations before we can consider your manuscript for acceptance. Your revisions should address the specific points made by each reviewer.

(1) A letter containing a detailed list of your responses to the review comments and a description of the changes you have made in the manuscript. Please note while forming your response, if your article is accepted, you may have the opportunity to make the peer review history publicly available. The record will include editor decision letters (with reviews) and your responses to reviewer comments. If eligible, we will contact you to opt in or out.

(2) Two versions of the manuscript: one with either highlights or tracked changes denoting where the text has been changed; the other a clean version (uploaded as the manuscript file).

Additionally, to enhance the reproducibility of your results, PLOS recommends that you deposit your laboratory protocols in protocols.io, where a protocol can be assigned its own identifier (DOI) such that it can be cited independently in the future. For instructions see http://journals.plos.org/plospathogens/s/submission-guidelines#loc-materials-and-methods

We hope to receive your revised manuscript within 60 days. If you anticipate any delay in its return, we ask that you let us know the expected resubmission date by replying to this email. Revised manuscripts received beyond 60 days may require evaluation and peer review similar to that applied to newly submitted manuscripts.

There is additional documentation related to this decision letter. To access the file(s), please click the link below. You may also login to the system and click the 'View Attachments' link in the Action column.

[LINK]

Sincerely,

David T. Evans

Associate Editor

PLOS Pathogens

Susan Ross

Section Editor

PLOS Pathogens

Kasturi Haldar

Editor-in-Chief

PLOS Pathogens

orcid.org/0000-0001-5065-158X

Michael Malim

Editor-in-Chief

PLOS Pathogens

orcid.org/0000-0002-7699-2064

Dear Molly,

I am sorry this has taken so long. The reviewers did the best they could, but were delayed both by the holiday season and the January 7th NIH AIDS review deadline. While the comments are generally positive, they have made some constructive suggestions that should be addressed to strengthen this work.

David

Reviewer's Responses to Questions

**Part I - Summary**

Reviewer #1: The manuscript describes results from a follow up study using the HIV-CRISPR screen to identify restriction factors targeting the N74D and P90A mutants of HIV-1 CA. TRIM34 was found to inhibit the N74D mutant, in a step that is prior to reverse transcription and in a process that is independent of CPSF6 binding but dependent on the presence of TRIM5α. The effect seems to be more pronounced in primary CD4+ T cells and TRIM34 also blocks SIVmac and SIVagm. The manuscript is well written and the findings provide important evidence for yet another capsid-dependent retrovirus restriction factor.

Reviewer #2: Ohainle et al. present interesting findings that provide some mechanistic insight into how TRIM34 inhibits HIV replication. While TRIM34 has previously been studyied in the context of lentiviruses, the authors show here that TRIM34 function requires TRIM5. Importantly, the authors take advantage of their HIV-CRISPR screening tool to generate these novel results in an unbiased manner. Overall, I feel that this manuscript is appropriate for PLOS Pathogens granted that some revisions are done.

Reviewer #3: In this manuscript, Ohainle et al. describe a novel interaction between human TRIM34 and HIV-1 capsid with the N74D mutation. They show that N74D HIV-1 and 2 SIV strains (but not WT HIV-1) are restricted by TRIM34 in both THP-1 cells and primary CD4+ T cells and monocyte-derived macrophages with or without type I IFN treatment. Restriction of N74D HIV-1 occurs at or before reverse transcription and requires the host restriction factor TRIM5a. In cells overexpressing tagged forms of TRIM5a and TRIM34, they can be colocalized with each other and are associated with capsid after infection with N74D HIV-1 but not WT HIV-1. Overall, the manuscript suggests that TRIM34 recognizes N74D CA (and SIV CA) in primary cells, leading to its previously described restriction. This finding could explain why N74 is highly conserved in primate lentiviruses.

**Part II – Major Issues: Key Experiments Required for Acceptance**

Reviewer #1: (No Response)

Reviewer #2: Major issues:

1. The authors should check to see how P90 affects restriction by TRIM34 in their ectopic TRIM34 expression system (Figure 2D).

2. Authors should perform the experiments in Figures 4A and 4B in the presence and absence of IFN (maybe they already have this?)

3. One noticeable absence from this paper is the study of non-human primate TRIM34. The authors indicate that TRIM34 is not rapidly-evolving in primates, but have they checked for overlapping function with human TRIM34? The authors have an experimental system in place to study ectopic TRIM34, so this would be an easy experiment to include. This would be a good addition to the paper because previous studies have already highlighted a role for TRIM34 in HIV restriction.

Reviewer #3: 1. The authors state that loss of CPSF6 binding is not sufficient for TRIM34 restriction as A77V HIV-1 is not restricted. However, this is an unusual mutant that is not restricted in general in macrophages. It would more convincing for the authors to evaluate other capsid mutants that do not bind to CPSF6 and are restricted in primary cells (i.e. N57A/S and/or A105T). Also, does TRIM34 restrict non-primate lentiviruses (i.e. FIV and EIAV)?

2. The authors initially identified TRIM34 restriction of N74D HIV-1 in IFN-treated THP-1 cells and state in Lines 322-325: "Our HIV-CRISPR screen demonstrates that the increased IFN-sensitivity of the HIV-1 N74D CA mutant [19] is not due to an increased sensitivity to the known capsid-targeting restriction factors TRIM5α or MxB (Figure 1B) but rather is due to restriction by a novel capsid-targeting restriction factor, TRIM34." However, other than this initial screen described in Figure 1 and data shown in Figure 2A, the remainder of the manuscript does not appear to include IFN-treated cells. In Figure 2, the level of restriction with or without IFN treatment is similar (2.2-fold vs. 2.0-fold, respectively). It is unclear how TRIM34 enhances IFN sensitivity, as this is not shown in the presented data.

3. CypA is discussed throughout the manuscript as being an important capsid-binding factor. Yet other than evaluation of P90A HIV-1 (and SIV), CypA is not directly addressed. This is surprising as 1) some of the co-authors of this manuscript recently published that CypA protects HIV-1 from TRIM5α and 2) Ambrose et al. showed that N74D HIV-1 is more sensitive to loss of CypA binding than WT HIV-1. Is TRIM34 restriction of N74D HIV-1 influenced by CypA?

4. There are some concerns regarding the imaging experiments in Figure 5. First, the data represent staining at 2h post-infection, which would be significantly later than capsid uncoating. Do the authors anticipate earlier binding of TRIM34/TRIM5α to N74D CA? Second, do the HeLa cells that express tagged TRIM34 and TRIM5α restrict N74D HIV-1? Finally, colocalization of N74D CA is higher with TRIM34 with half of them not associated with TRIM5α. Does this suggest a sequential interaction of the 2 host proteins with N74D capsid?

**Part III – Minor Issues: Editorial and Data Presentation Modifications**

Reviewer #1: The TRIM34 restriction is only seen in lab generated mutants of HIV-1 capsid and not wild type. Unless these mutations occur naturally, the title claiming that TRIM34 restricts HIV-1 is a little misleading. Perhaps “primate lentiviral capsids” will be more appropriate.

Given that the restriction is stronger against the SIVs, it would seem that TRIM34 could work with TRIM5α to prevent cross-species transmission of primate lentiviruses, making the SIV results as important, if not more, than the weak effects seen with the artificial HIV-1 mutants. The authors have hinted at this in the penultimate paragraph on page 11 but the point could be emphasised more strongly.

TRIM34 scored as highly as the IFN pathway genes in the HIV-CRISPR screen. Yet, its effects are weak in subsequent experiments. Given that the readout of the HIV-CRISPR screen is from viruses that are released from infected cells, this would identify processes from early as well as late stages of the life cycle. However, the subsequent experiments measured percentage of infected cells after 48 hours, which is probably the result of a single cycle of infection, hence only early stages of the life cycle. If TRIM34 works with TRIM5α, which has been shown to also act as an immune sensor, the possibility that it could trigger the innate immune pathways cannot be discounted. This could activate genes that might act on the late stages of the viral life cycle and would not be picked up looking at infected cells from a single cycle infection. Indeed, N74D has been previously reported to trigger innate sensors due to lack of Cpsf6 binding. However, this will have to be reinterpreted in the light of the data from Fig 3A and 3B.

Other points:

Typos: Line 45 on page 2, “… the HIV-1 capsid protein is exposed to an array of …”

Line 150 on page 5, “However, unlike T TRIM5α ,…” T shouldn’t be there.

Fig 2A and 2B. The legend describes the assay as intracellular P24gag+ staining but the y-axis is labelled as %GFP+ cells. Is the readout p24gag+ or GFP+?

Fig 2C and Fig 3B. These are very similar experiments involving CD4+ T cells. However, the infectivities of WT and N74D viruses are different (5% GFP+ for WT, 1% GFP+ for N74D/TRIM34KO in Fig 2C; 1% GFP+ for WT, 4%GFP+ for N74D/TRIM34 KO in Fig3B). The effect of TRIM34KO also varies. (8x in Fig 2C and 4.2x in Fig 3B). Is it amount of input virus the same or normalized in anyway? The variation of 2-fold is of concern since some of the TRIM34 effects described (eg Fig 2A and 2B) are 2-fold.

Fig 2E. The significance of the graphs showing Neviparin treatments were not described in the text.

Reviewer #2: Minor issues:

1. Author summary: viruses don’t adapt to host genes, but rather host proteins. And HIV-1 does not interact with human genes (at least not in the sense that the authors are trying to describe). HIV-1 interacts with gene products (proteins).

2. Line 119: remove “are enriched in the viral supernatant.”

3. Fonts used to label different parts of Figure panels, including the titles, should be identical (I suggest Arial). For example, in Figure 2, “control” and “TRIM34” are Calibri, or something like that. This appears to be a common problem throughout all figures.

4. Figure 2: raw data (%GFP+ cells) makes an appearance in Figure 2. I suggest that some raw data is added to Figure 1 such that the effect of STAT, IFNAR, IRF9, and TRIM34 KO can be seen. It would also be nice to see the extent to which IFN inhibits LAI infection in THP1 here.

5. Can the authors discuss why TRIM34 KO rescues WT and N74D virus to nearly similar extents in the absence of IFN?

6. Line 187: what do the authors mean by a “knockout allele?”

7. Line 230: this statement is confusing. Actually, due to ongoing genetic conflicts between host and pathogens, restriction factors from one species may not restrict viruses adapted to other species (because virus capsids co-evolve with host-specific capsid-targeting factors). I suggest removing this line entirely as it is not required to ask whether TRIM34 acts broadly.

8. It is interesting to show and discuss the data in Figure 1 featuring both P90A and N74D. And the results from that Figure 1 justify a focus on N74D in subsequent figures. However, I would remove mention of P90A from the abstract as it is not the main focus of the paper and a recent paper has characterized the relationship between TRIM5 and this CypA binding deficient mutant.

9. I think that more discussion is warranted on the following topic: does TRIM5 need TRIM34 in order to restrict HIV (WT or N74D)? Why or why not? The authors only focus the discussion on how TRIM34 needs TRIM5alpha in order to restrict N74D. The title states how “TRIM34 works with TRIM5.” This word choice is vague, and it begs the question of whether some or all activities of TRIM5 are dependent on TRIM34. This title may also give the impression that TRIM34 and TRIM5 work in an additive (independent) fashion. An alternative title could be: “TRIM34 restricts HIV and SIV capsids in a TRIM5-dependent manner.”

10. The authors should include a mention of the data that TRIM34 has different specificity than TRIM5 (it inhibits SIVmac) in the Discussion. Can they speculate as to why TRIM34+TRIM5 may restrict SIVmac CA but TRIM5 alone does not?

11. Line 386: the language used here is confusing. Do references 43 and 44 conflict with one another? Does N74D exhibit slower uncoating kinetics in one paper but not the other? Similarly, at line 394, is it implied that N74D uncoats and transits more slowly than WT? Please clarify.

Reviewer #3: 1. Line 45: the word "an" should precede "array of capsid-targeting"

2. Lines 103-104: P90A is not CypA-deficient but rather CypA binding-deficient. Similarly, N74D is not CPSF6-deficient but deficient in binding to CPSF6.

3. Line 133: The preceding sentence describes "ISGs, including MXB (MX2), IFITM1, TETHERIN (BST2) and TRIM5." Only IFITM1 and TETHERIN is in cyan in Figure 1; TRIM5 and MXB are yellow and dark blue, respectively.

4. Line 327: should be "G89V" instead of "G98V"

PLOS authors have the option to publish the peer review history of their article (what does this mean?). If published, this will include your full peer review and any attached files.

Reviewer #1: No

Reviewer #2: Yes: Alex Compton

Reviewer #3: No

---

## [Editor Report · Decision Letter 1]

29 Mar 2020

Dear Dr. Ohainle,

We are pleased to inform you that your manuscript 'TRIM34 restricts HIV-1 and SIV capsids in a TRIM5α-dependent manner' has been provisionally accepted for publication in PLOS Pathogens.

Best regards,

David T. Evans

Associate Editor

PLOS Pathogens

Susan Ross

Section Editor

PLOS Pathogens

Kasturi Haldar

Editor-in-Chief

PLOS Pathogens

orcid.org/0000-0001-5065-158X

Michael Malim

Editor-in-Chief

PLOS Pathogens

orcid.org/0000-0002-7699-2064
---

## [Editor Report · Acceptance letter]

6 Apr 2020

Dear Dr. Ohainle,

We are delighted to inform you that your manuscript, "TRIM34 restricts HIV-1 and SIV capsids in a TRIM5α-dependent manner," has been formally accepted for publication in PLOS Pathogens.

Best regards,

Kasturi Haldar

Editor-in-Chief

PLOS Pathogens

orcid.org/0000-0001-5065-158X

Michael Malim

Editor-in-Chief

PLOS Pathogens

orcid.org/0000-0002-7699-2064